# High-Condensed Tannin Diet and Transportation Stress in Goats: Effects on Physiological Responses, Gut Microbial Counts and Meat Quality

**DOI:** 10.3390/ani11102857

**Published:** 2021-09-30

**Authors:** Phaneendra Batchu, Toni Hazard, Jung H. Lee, Thomas H. Terrill, Brou Kouakou, Govind Kannan

**Affiliations:** Agricultural Research Station, Fort Valley State University, Fort Valley, GA 31030, USA; pbatchu@wildcat.fvsu.edu (P.B.); hazard74922@gmail.com (T.H.); leej@fvsu.edu (J.H.L.); terrillt@fvsu.edu (T.H.T.); kouakoub@fvsu.edu (B.K.)

**Keywords:** catecholamines, goats, meat quality, rumen bacteria, sericea lespedeza, stress

## Abstract

**Simple Summary:**

Preslaughter management of goats can affect their welfare and productivity, as well as the resultant product quality. Inclusion of high-condensed tannin ingredients in the diet and minimizing stress can improve meat quality and safety in goats. This study was conducted to determine the effects of transportation stress and sericea lespedeza hay diet on physiological responses, gastrointestinal microbial populations, and meat quality in Spanish goats. Although transportation stress influenced physiological responses, it did not influence muscle pH, which is an important variable that affects the eating quality of meat. Sericea improved anti-inflammatory property and decreased microbial counts in the rumen and rectum of goats. There is evidence that feeding high-condensed tannin diet could be beneficial in meat goat production.

**Abstract:**

Feeding condensed tannin (CT)-containing diets such as sericea lespedeza (*Lespedeza cuneata*) and reducing stress have been reported to improve meat quality and food safety in goats. In a completely randomized design with split-plot, thirty-six uncastrated male Spanish goats were assigned to 3 dietary treatments (n = 12/treatment): ground ‘Serala’ sericea lespedeza hay (SER), bermudagrass (*Cynodon dactylon*) hay (BG), or bermudagrass hay—dewormed goats (BG-DW; Control) at 75% of intake, with a corn-based supplementation (25%) for 8 weeks. Prior to slaughter, goats were either transported for 90 min to impose stress or held in pens. Basophil counts were lower (*p* < 0.01) in the SER group compared to BG or BG-DW groups suggesting a better anti-inflammatory capacity due to polyphenols in the SER diet. Compared to BG-DW group, cortisol level was higher (*p* < 0.05) and norepinephrine was lower (*p* < 0.05) in the SER group. The SER group had the lowest aerobic plate counts (APC) in both rumen and rectum (*p* < 0.01). *Longissimus dorsi* muscle initial pH was not affected by diet or stress. Feeding sericea hay to goats may have beneficial effects, such as enhanced antioxidant and anti-inflammatory properties during stress and reduced gut microbial counts, without changing meat quality characteristics.

## 1. Introduction

Spanish goats are commonly raised for meat production in the southeastern US. Goats selectively browse on high quality and more nutritious parts of plants, including coarse weeds, grasses, and some legumes.

Severe preslaughter stress related to transportation has been reported as one of the major factors affecting meat quality in small ruminants [1]. Evidence from previous studies suggests that transportation stress impacts adrenal cortical activity, immune response, meat quality, and body weight in livestock [2,3]. An increase in cortisol and glucose levels were reported in goats that were transported for 2 h [4]. Nwe et al. [5] reported that in Japanese native goats, plasma cortisol concentrations increased within 30 min after beginning of transport and reached a peak concentration at 1 h. In Omani goats subjected to transportation stress, increased concentrations of plasma cortisol, epinephrine, norepinephrine, and dopamine were observed [6].

Feeding diets rich in condensed tannins (CT) have been used to alleviate the negative impact of transportation stress on carcass and meat quality [7]. Supplementation with CT in ruminant diets can influence beneficial fatty acid content of meat such as linolenic acid, vaccenic acid and rumenic acid [8]. Moreover, the tannin-protein complexes that are formed when tannin-rich diets are fed to ruminants are reported to be harmful to gastrointestinal nematodes [9,10]. Sericea lespedeza (*Lespedeza cuneata*) is a perennial, drought tolerant legume that grows well on acidic soils with low fertility [11] and contains high concentration of CT. Tannins form complexes with protein, polysaccharides, and minerals when the cell integrity of the plants is disrupted via chewing, drying, and/or grinding [12]. The ability to combine with such molecules can alter ruminant digestion by inhibiting the growth of rumen microorganisms [13]. Feeding ground sericea reduced bacterial counts in goat feces compared to a bermudagrass hay diet [14].

Sericea lespedeza is known to have antioxidant properties that reduce free radical production [15]. In vitro studies using *Lespedeza virgata* have revealed presence of natural antioxidant properties in this species [16]. Antioxidants are substances that can prevent or slow damage to cells caused by free radicals and are essential in reducing the formation of O_2_ radicals, such as superoxide, and protecting the immune system [17]. Dietary antioxidant supplements have been reported to increase the antioxidant status of muscles and improve shelf-life of goat meat [18]. However, the effects of a high-CT diet such as sericea lespedeza on reducing microbial populations in the gut and improving meat quality and antioxidant status, particularly when goats are exposed to preslaughter stress such as transportation are not well documented.

There are no data available on plasma catecholamine responses to ‘Serala’ sericea lespedeza diet and stress in goats. The objectives of this experiment are to determine the effects of a high-CT diet (sericea lespedeza hay), preslaughter stress, and their interactions on plasma hormones and metabolites, meat quality characteristics, and gut microbial counts in Spanish goats.

## 2. Materials and Methods

### 2.1. Animals and Diets

This research protocol was reviewed and approved by the Fort Valley State University’s Animal Care and Use Committee (Approval # F-R-01-2019) following the ADSA-ASAS-PSA Guide for Care and Use of Agricultural Animals in Research and Teaching [19] prior to starting the experiment. In a Completely Randomized Design, thirty-six uncastrated male Spanish goats (8-month old; body weight 26.1 ± 0.16 kg) were assigned to 3 dietary treatments: ground ‘Serala’ sericea lespedeza hay (SER), bermudagrass hay (BG), or bermudagrass hay—dewormed goats (BG-DW) at 75% of intake, with a corn-based supplementation (25%) for 8 weeks. The SER group was compared with the BG group in this study since goats in the southeastern US are commonly raised on a diet comprising of mixed pastures primarily consisting of bermudagrass, a concentrate supplement in some cases, and ad libitum bermudagrass hay. The BG-DW group was maintained as a control since meat goats raised in the southeastern US are also typically infected to some degree with gastrointestinal parasites. As SER was expected to have an antioxidant combined with anti-parasitic effect on animals, the treatment structure employed helped evaluate if SER had an overall beneficial effect on goats. The diets were formulated (Table 1) and mixed at our facilities and samples of hay and concentrates collected at the beginning, middle and end of the experiment were pooled and analyzed for nutrient content at a commercial laboratory (Dairy One Forage Laboratory, Ithaca, NY). The estimates of diet nutrient composition (hay + concentrate) were then calculated and the CP, ADF, NDF, and TDN% were 18.3%, 27.4%, 54.5%, and 63.5%, respectively, in the BG diet, and 19.3%, 46.5%, 59.0%, and 60.5%, respectively, in the SER diet. The ‘Serala’ sericea lespedeza hay samples analyzed had a total CT content of 5.1% and the bermudagrass hay samples had virtually no CT present. The ingredients used in the concentrate portion of the diet have zero to negligible CT content. Therefore, the SER group received high CT from the diet and the other two groups received virtually no CT from the diet. Animal is the experimental unit since goats were kept in individual metal grill pens (1.5 × 1.5 m) during the feeding trial. At the end of the trial, the animals were weighed and transported to the processing facility and held overnight without feed in holding pens prior to slaughter. On the day of slaughter, half the goats were transported (TS) for 90 min to impose stress (stress treatment, ST) using a livestock trailer (5.0 × 2.3 m; floor space 1.27 m^2^/goat) and the remaining goats were held in pens (NTS) before slaughtering in two replicates (2 different days). On the days of the trial, the high temperature recorded at the research station was 15.6 ± 1.0 °C, the low temperature was 5.9 ± 1.0 °C, and the relative humidity was 73 ± 2%. Animals were slaughtered using humane procedures.

### 2.2. Blood Sampling and Analysis

Blood samples were collected by trained personnel as quickly as possible after the goats were caught in order to avoid confounding of the effect of blood sampling. All efforts were made not to agitate the goats, including avoiding loud noise and rough handling and minimizing metal sounds. Blood samples were collected by jugular venipuncture into disposable vacutainer tubes containing 81 µL of 15% EDTA solution. Each animal was sampled four times (two from each side) during the 90 min treatment period (0, 30, 60, and 90 min) with approx. 5 mL of blood collected each time. The blood tubes were placed on ice until blood smears were made and plasma separated. The samples were centrifuged at 1000× *g* for 20 min, plasma pipetted into different aliquots, and then stored at −80 °C until analysis.

Biogenic amine assay was conducted by The Metabolomics Innovation Center (TMIC), University of Alberta, Edmonton, AB Canada using the methodology described by Zheng et al. [20]. Plasma samples were thawed on ice, in the dark, before use. For plasma analysis, 10 μL of the internal standard (ISTD) mixture solution and 25 μL of samples (PBS for “zero” samples, calibration curve standards, and plasma samples) were pipetted directly onto the center of each spot in a 96-well filter plate. The whole plate was evaporated under nitrogen flow to dryness for 45 min. Then, 50 μL of phenylisothiocyanate (PITC) derivatization solution (which derivatizes free amines) was added to each well, and the reaction kept at room temperature for 20 min to allow the derivatization reaction to finish. After completion of the reaction, the samples were again dried in the nitrogen evaporator for 90 min to remove any liquid, especially the excess PITC solution, followed by the addition of 75 μL of methanol containing 5 mM ammonium acetate. The 96 well plate was then covered and shaken at 450 rpm for 30 min at room temperature, and then spun in a centrifuge for 15 min at 1200 rpm. To each well of the collection plate, 75 μL of water was added and mixed thoroughly, and then 40 μL was injected into an HPLC-equipped QTRAP 4000 mass spectrometer for LC-MS/MS analysis. Plasma serotonin analysis was also conducted at the TMIC using a combination of direct injection mass spectrometry with a reverse-phase LC-MS/MS custom assay.

Plasma cortisol levels were measured by using Cortisol ELISA Kit (Abnova Corporation, Taipei city, Taiwan), which is based on a widely used immunoassay technique. A plasma sample containing an unknown amount of cortisol to be assayed (unlabeled antigen) is added to a standard amount of a labeled derivative of the same substance (labeled antigen). The labeled and unlabeled antigens are then allowed to compete for high affinity binding sites on a limited number of antibodies coated onto the plate. After washing away the free antigen, the amount of labeled antigen in the sample is reversibly proportional to the concentration of the unlabeled antigen. The actual concentrations in unknown samples are obtained by means of a standard curve based on known concentrations of unlabeled antigen analyzed in parallel with the unknowns. In this kit, an enzyme label is used. The biospecific reaction takes place during 1 h incubation. After washing, substrate solution is added, and the enzyme allowed to react for a fixed time before the reaction is terminated. Absorbencies are measured at 450 nm using an ELISA plate reader.

### 2.3. Differential Leukocyte Counts

Two blood smears were made from each 0 min and 90 min blood sample on microscopic slides. The blood smears were dried at room temperature and stained with JorVet dip quick stain. Neutrophils, lymphocytes, basophils, eosinophils, and monocytes were identified under the microscope using 100/1.25 oil immersion objective, and counted (100 cells per slide) using the straight edge method described by Schalm et al. [21].

### 2.4. Rumen and Rectal Sampling and Microbial Analysis

After evisceration, rumen content and rectal pH were determined using a pH meter, and rumen fluids were collected and processed for VFA analysis. Rumen and rectal digesta samples were collected separately into sterile containers, placed in an icebox, and transported to the laboratory for microbial analysis. A 10 g sample of rumen or rectal contents was transferred into a sterilized stomacher bag (gamma-irradiated polyethylene bags, Fisher Scientific, Pittsburgh, PA, USA), and 90 mL of 0.1% sterile buffered peptone water (Difco Laboratories, Detroit, MI, USA) was added. The content in the bag was pummeled in a stomacher lab blender (Seward Model 400, Tekmar Co., Cincinnati, OH, USA) for 2 min. Serial dilutions were prepared with 0.1% sterile buffered peptone water (Difco Laboratories). The 3M™ Petrifilim plate techniques were used to enumerate microbial loads on rumen and rectum samples as recommended by the manufacturer (3M™ Microbiology Products, 1999). Appropriate sample dilutions were inoculated on Petrifilm plates (3M™ Microbiology Products, St. Paul, MN, USA) to determine *E. coli* (3M™ Petrifilm™ *E. coli*/coliform Count Plates). Appropriate dilutions were also inoculated on trypticase soy agar (TSA) plates to determine total plate counts. Colonies were counted after 24 h incubation at 35 °C for *E. coli* and total coliform counts, and after 48 h incubation for aerobic plate counts.

### 2.5. Volatile Fatty Acid (VFA) Analysis

After slaughter, rumen acidity (pH) was determined using the Beckman Coulter Φ ^®^400 Series pH meter (Beckman Coulter Inc., Fullerton, CA, USA). The rumen content samples from each animal was strained through eight layers of cheesecloth and collected in 50 mL plastic tubes. To halt any biological processes, each filtrate sample was acidified with 1%/v of 7.2 N sulfuric acid. Samples were frozen at −80 °C for ammonia nitrogen and volatile fatty acid analysis. Rumen fluid samples were thawed overnight (12 h ± 30 min) at 4 °C. Samples were centrifuged (Thermo Fisher Sorvall Lynx 6000; Thermo Fisher Scientific, Waltham, MA, USA) at 10,000× *g* for 10 min. A sub-sample of the supernatant was removed for rumen ammonia analysis using the phenol-hypochlorite assay adapted from Broderick and Kang [22]. Absorbance values were measured using the Synergy HTX microplate reader (Bio-Tek—Winooski, VT, USA). The remaining supernatant was centrifuged again at 10,000× *g* for 10 min. and stored for volatile fatty acid analysis. Volatile fatty acids were determined using a Thermo Electron Gas Chromatography unit (Thermo Electron Corporation, Waltham, MA, USA).

### 2.6. Meat Quality Analysis

Initial muscle pH was determined within 15 min (initial pH, immediately after skinning) using a portable pH meter with a penetrating probe (Pakton^®^ Model OKPH1000N, Vernon Hills, IL, USA). The probe was inserted directly into the *Longissimus dorsi* muscle to determine pH. The carcass was stored at 2 °C for 24 h before determination of final pH and fabrication. After 24 h of storage, each carcass was weighed, and 2.5 cm thick loin/rib chops were collected for analysis of meat quality characteristics. Four loin/rib chops from each carcass were allowed to bloom for 40 min and used to measure color values (L*, a*, b*) using a MiniScan^®^ XE Plus colorimeter (Serial No. 6130; Hunter Lab, Reston, VA, USA). The color values were determined directly on the cut surface of chops. Four chops from each carcass were vacuum packed and stored at −20 °C for cooking loss and texture analysis. The cuts were thawed at 4 °C, weighed, placed on aluminum pans, and covered with aluminum foil. The chops were cooked in a convection oven (Maytag model, Redwood, CA, USA) to an internal temperature of 71 °C. The samples were then allowed to come to room temperature before measuring the final weight and calculating cooking losses. The Warner-Bratzler shear force (WBSF) values were assessed using a TA-XT2 Texture Analyzer (Texture Technologies Corp., Scarsdale, NY, USA), with the samples (1 cm diameter cores) being sheared at right angles to the orientation of muscle fibers using a Warner-Bratzler shear attachment (Texture Technologies Corp., Scarsdale, NY, USA). The instrument was set with a 25 kg load cell and a crosshead speed of 200 mm/min. Cooking loss calculation and WBSF value determination were done according to the procedures described by Kannan et al. [23].

### 2.7. Statistical Analysis

Blood data were first examined for normality and homogeneity of variance by plotting residual vs. predicted values, Levene’s Test, and Shapiro–Wilk’s Test. Repeated Measures Analysis was then conducted using GLM procedures in SAS. The sphericity test was performed to check if orthogonal components were uncorrelated and had equal variances. When appropriate, the Greenhouse-Geisser Test probability value was taken into account for within-subject effects. Because the levels of time represented a quantitative factor, a separate Repeated Measures Analysis was conducted using polynomial contrasts when the Time effect was significant at *p* < 0.05 to assess the trends of plasma hormone and biogenic amine concentrations in TS and NTS groups. In addition, Pearson correlation analysis was conducted to study the association between cortisol and biogenic amines. Meat quality characteristics, rumen and rectal pH and microbial counts, and rumen VFA data were analyzed as a split-unit design using the GLM procedures in SAS with Diet as the whole-unit factor and ST as the split-unit factor with error terms specified. Cortisol and biogenic amine data were transformed to log scale to meet the assumptions of ANOVA; however, the means and standard errors were back transformed and presented. When significant by ANOVA at *p* < 0.05, the means were separated using the least significant difference (LSD) test.

## 3. Results

### 3.1. Plasma Cortisol Concentrations

Plasma cortisol was significantly lower (*p* < 0.05) in the BG-DW group compared to BG or SER diet groups (*p* < 0.05; Figure 1A). Plasma cortisol concentrations were higher in TS goats compared to NTS goats (*p* < 0.01), and the values did not change over time in the NTS goats, while the values increased over time in the TS goats (ST × Time, *p* < 0.01, Figure 1B). Application of polynomial contrasts revealed that the lines followed a significant linear trend (*p* < 0.01; Figure 1C).

### 3.2. Plasma Biogenic Amine Concentrations

Plasma epinephrine concentrations were not affected by any of the factors studied (Figure 2A). Plasma norepinephrine concentrations were low (*p* < 0.05) in the SER diet group, high in the BG-DW group, and intermediate in the BG group (Figure 2B). The changes in plasma norepinephrine concentrations over Time were different among the different Diet groups and the Time patterns were not consistent between the NTS and TS groups (Diet × ST × Time, *p* < 0.01; Figure 2C). For instance, the norepinephrine concentrations in BG-DW animals peaked at 90 min in the NTS group, while the concentrations in these animals peaked at 60 min in the TS group. Plasma dopamine concentrations were not affected by any of the factors studied; however, there was a tendency toward a ST × Time effect (*p* = 0.07; Figure 3A). There was also an ST × Time interaction effect on seratonin concetrations (*p* < 0.05; Figure 3B).

Plasma metanephrine and normetanephrine concentrations were higher (*p* < 0.01) in the TS group compared to NTS group. Time (*p* < 0.01) and ST × Time (*p* < 0.01) also influenced metanephrine concentrations, with the levels remaining low and not changing over time in the NTS group, while the levels significantly increasing at 30 min after transportation and remaining at that level throughout the transportation duration in the TS group (Figure 4A). Polynomial contrasts showed that the lines followed a cubic trend (*p* < 0.01; Figure 4B) over Time. In addition to the significant Time (*p* < 0.01) and ST × Time (*p* < 0.01) effects, Diet × ST was also significant (*p* < 0.01) for normetanephrine concentrations (Figure 4C), since the Diet effect was more prominent in the TS group compared to the NTS group. The lines defining the Time relationship followed a quadratic trend (*p* < 0.01; Figure 4D) based on polynomial contrast analysis.

There was a tendency for plasma tyramine concentrations to be higher in the BG-DW group compared to the other Diet groups (Diet, *p* = 0.07; Figure 5A). Overall tyramine concentrations in the TS group were low at 0 min, and increased and peaked at 60 min before decreasing at 90 min of transportation; however, this trend was not seen in the NTS group (ST × Time, *p* < 0.05; Time, *p* < 0.05; Figure 5B). The concentrations over Time followed a linear trend (*p* < 0.01; Figure 5C). Overall plasma phenylethylamine levels decreased gradually and reached the lowest level at 60 min and increased again at 90 min (Time, *p* < 0.01; Figure 6A), and there was a significant cubic trend (*p* < 0.05; Figure 6B) over Time. Plasma 5-methoxytryptamine concentrations were not affected by any of the factors studied (Figure 6C).

Pearson correlation analysis (Table 2) between cortisol and selected biogenic amines revealed that the correlation coefficients were significant for cortisol vs. metanephrine (*p* < 0.01) and cortisol vs. normetanephrine (*p* < 0.01) analyses, except in the SER group. A correlation matrix is not included to avoid presenting spurious correlations.

### 3.3. Differential Leukocyte Counts

The lymphocyte counts were higher in the NTS groups compared to the TS groups (*p* < 0.01), and the counts were higher at 0 min compared to 90 min (*p* < 0.01). The neutrophil counts were higher in the TS groups compared to the NTS groups (*p* < 0.01), and the counts were higher at 90 min compared to 0 min (*p* < 0.01). The ST × Time interaction was also significant for both lymphocyte (*p* < 0.01) and neutrophil (*p* < 0.01) counts, with numbers not changing with Time in the NTS group, while the lymphocyte counts decreasing and neutrophil counts increasing at 90 min in the TS group (Table 3). The monocyte counts were high in BG, low in BG-DW, and intermediate in SER group (*p* < 0.05). Basophil counts were lower (*p* < 0.01) in SER group compared to BG or BG-DW groups (Table 3).

### 3.4. Gut Microbial Counts and VFA

Diet or ST did not have a significant effect on rumen and rectal pH values and *E. coli* (Table 4). Diet × ST interaction was significant for rectal pH (*p* < 0.05). The SER group had the lowest aerobic plate counts (APC) in both rumen and rectum (*p* < 0.01). Rumen APC were 4.74, 6.74, and 6.95 (SEM = 0.420) and rectal APC counts were 6.53, 7.26, and 6.75 (SEM = 0.138) log10 CFU/g in SER, BG, and BG-DW groups, respectively. Diet did not affect VFA; however, acetic acid, isobutyrate and valerate percentages were influenced by ST (*p* < 0.01; Figure 7). Acetic acid concentration was lower (*p* < 0.01) and isobutyrate and valerate concentrations were higher (*p* < 0.01) in TS goats compared to NTS goats.

### 3.5. Meat Quality Characteristics

Muscle initial and final pH values was not affected by Diet or ST (Table 5). The a* values (redness) were lower in SER compared with BG or BG-DW (*p* < 0.01). Meat color values were not affected by ST and cooking loss and WBSF were also not influenced by ST or Diet (Table 5).

## 4. Discussion

Meat goat management practices that increase animal stress may alter the normal course of conversion of muscle to meat leading to inferior meat quality [1]. Stress in animals can negatively affect animal growth, weight gain, immune function, and physiological status that in turn can lead to poorer health, carcass yield, and economic returns [23].

### 4.1. Plasma Cortisol Concentrations

Earlier studies have shown that diet does not affect plasma cortisol concentrations [24], although feed deprivation can increase cortisol concentrations [25]. In the present study, however, plasma cortisol was significantly lower in the BG-DW group compared to BG or SER diet groups. This effect in BG-DW animals is likely due to lower gastrointestinal parasite load rather than diet itself. The BG-DW animals were maintained as dewormed controls in this study. An earlier study [26] revealed that parasite load did not significantly affect plasma cortisol concentrations in goats. Although there is no direct relationship between severity of gastrointestinal parasite infection and plasma cortisol concentrations in these studies, it is likely that the general nutritional and health status of goats devoid of parasites is better and therefore the animals are in a better position to cope up with the negative effects of stress. Stress results in an increase in circulating glucocorticoids during the restorative phase of the fight or flight reflex that helps animals replenish glycogen reserves initially depleted or decreased in muscles due to the release of catecholamines during the alarm phase. Plasma cortisol concentrations in the present study were higher in TS goats compared to NTS goats, and the values did not change over time in the NTS goats, while the values increased over time in the TS goats. These results are consistent with previous reports on the effects of transportation stress on cortisol concentrations in goats [1,25]. Loading meat goats onto transport trailer, and a 2 ½-hour transportation combined with an 18 h feed deprivation all have been reported to increase stress as indicated by plasma cortisol concentrations in goats [25]. Since feed was withdrawn overnight in the present experiment, feed deprivation time also increased with transportation time in the TS group and there was a linear increase in cortisol concentrations over time, while such an effect was absent in the NTS group. When goats are feed deprived without transportation, cortisol concentrations do not increase [27].

### 4.2. Plasma Biogenic Amine Concentrations

Catecholamines are synthesized from phenylalanine or derived from tyrosine, an amino acid from dietary sources [28]. Tyrosine is also derived from hydroxylation of phenylalanine by the enzyme phenylalanine hydroxylase. The biosynthesis of catecholamines involves conversion of tyrosine to L-DOPA, to dopamine by catecholamine-secreting cells, and then to norepinephrine and epinephrine [29]. Catecholamines are produced primarily in the adrenal medulla by the chromaffin cells and the postganglionic fibers of the sympathetic nervous system. Among other factors, psychological and environmental stress, low blood glucose, and a stimulation of the sympathetic nervous system can result in elevated circulating catecholamine concentrations. Epinephrine concentrations generally increase in response to stress, and elevated epinephrine concentrations due to transportation stress in goats is previously documented [5]. However, in the present study epinephrine concentrations were not affected by ST or Time. Plasma norepinephrine concentrations were lower in the SER diet group compared to BG-DW or BG groups. Generally, polyphenols such as catechins inhibit the enzyme catechol-O-methyl transferase, which degrades norepinephrine [30]. If the CT present in sericea lespedeza has had a similar effect, the norepinephrine concentrations would have been higher in SER group than the other diet groups, which was not the case. A plausible explanation for lower plasma norepinephrine concentrations in the SER group is lower sympathetic nervous system activity. There is increasing evidence that dietary polyphenols can reduce sympathetic nervous system over activity, in addition to counteracting oxidative stress [31]. The lack of significant transportation time main effect on norepinephrine concentrations was in agreement with the results observed by Nwe et al. [5] in Japanese native Tokara goats. Circulating levels of norepinephrine depend on rate of tissue clearance, reuptake processes, receptor sensitivity, and regional sympathetic nervous function, in addition to secretion from nerve terminals [32]. The effects of diet and interactions on plasma epinephrine and norepinephrine concentrations were not the same, which could be explained by the fact that the sympathetic nervous system and the peripheral catecholamine systems are governed by separate regulatory mechanisms [33].

Several peripheral catecholaminergic systems exist that can be differently controlled by different stressors [34]. Plasma dopamine is primarily from sympathetic noradrenergic nerves as it is present only in small amounts in the adrenal gland compared to epinephrine and norepinephrine [34]. Dopamine concentrations were not affected by Diet, ST, or Time in the current study. Serotonin, a monoamine neurotransmitter known to be involved in fear responses and anxiety in mammals, was not affected by Diet in this experiment. Although serotonin responses over time were different in the two ST groups, there was no discernable pattern.

Metanephrine and normetanephrine are O-methylated metabolites of catecholamines, epinephrine and normetanephrine. In the present study, plasma metanephrine and normetanephrine concentrations were higher in the TS group compared to NTS group. These responses were not similar to those of plasma epinephrine and norepinephrine concentrations in this study. This was probably due to the fact that epinephrine has a half-life of a few minutes and that it is quickly broken down to products such as metanephrine and normetanephrine. Metanephrine and normetanephrine concentrations remained low and did not change over transportation time in the NTS group, while the levels significantly increased at 30 min after transportation and remained at that level throughout the transportation duration in the TS group. In addition, the diet effect on normetanephrine concentrations was more prominent in the TS group compared to the NTS group explaining a Diet × ST interaction effect. It appears that metanephrine and normetanephrine may be better measures to study the effects of Diet, ST, Time and their interactions than epinephrine and norepinephrine in goats. Since plasma cortisol concentrations are regarded as good indicators of stress levels in goats, a significant correlation between cortisol and metanephrine concentrations further strengthens the possibility that these metabolites of catecholamines could be more reliable indicators of stress than the other biogenic amines. Increases in plasma metanephrine and normetanephrine concentrations have been previously recorded in response to a 30 min restraining stress in rabbits [35].

In the present study, plasma tyramine concentrations in the TS group were low at 0 min and increased and peaked at 60 min before decreasing at 90 min of transportation; however, this trend was not seen in the NTS group. This response is not similar to that seen for dopamine concentrations, although both tyramine and dopamine are synthesized from the common immediate precursor, tyrosine. Tyramine is synthesized from decarboxylation of tyrosine by the enzyme tyrosine decarboxylase [36]. Although phenylethylamine levels are greatly affected by diurnal variations in animals, increase in this trace amine compound has been reported in rats in response to stress [37]. In our study, plasma phenylethylamine levels decreased gradually and reached the lowest level at 60 min and increased again at 90 min. It is not clear if the mechanisms of synthesis and release of phenylethylamine are different between rats and small ruminants.

### 4.3. Differential Leukocyte Counts

A higher neutrophil to lymphocyte ratio has been reported in goats fed a growth plus maintenance diet compared to those fed a twice growth plus maintenance diet, indicating the effect of nutrition level [38]. The authors also found a nutrition-parasite load interaction effect on neutrophil and lymphocyte counts. Although parasite load was not reported in this study, it is clear that there was no significant effect of diet on neutrophils and lymphocyte counts. However, stress increased neutrophil counts and decreased lymphocyte counts. The neutrophil counts were higher in the TS groups compared to the NTS groups, and the counts were higher at 90 min compared to 0 min. The cell counts did not change with time in the NTS group, while the lymphocyte counts decreased, and neutrophil counts increased at 90 min in the TS group. These results are in agreement with the earlier studies conducted on the effects of transportation stress on differential leukocyte counts in goats [25].

Basophil counts were lower in the SER group compared to BG or BG-DW groups. Basophils are a type of granulocytes and they play an important role in immune function, and thus the counts can increase due to inflammation [39]. The lower basophil counts can suggest anti-inflammatory properties of the polyphenols present in the SER diet compared to the BG diet. An earlier report indicated that nutrition level had a significant effect on basophil counts and immature white cells in goats [38]. The monocyte counts were higher in the BG compared to BG-DW and SER groups. Monocyte counts can increase in response to stress and inflammation. The lower monocyte counts in SER and BG-DW groups suggest a better capacity by goats to combat the negative effects of stress because of either the beneficial effects of high CT content, low gastrointestinal parasite load, or both. Eosinophil counts were not influenced by any of the factors, although there have been reports that eosinophil counts decreased after the beginning of transportation in goats [5]. However, the results of the present experiment are in agreement with those of other authors [40] who reported that nutrition did not have any effect of eosinophil counts in creole goat kids.

### 4.4. Gut Microbial Counts and VFA

An earlier study with sheep and goats showed that *E. coli*, coliform, *Enterobacteriaceae*, and total plate counts were more associated with pH in the colon than in the rumen [41]. In the current study, the rectal pH values were lower in TS than in NTS goats in BG and BG-DW diet groups, and the pH values were higher in TS than in NTS goats in the SER diet group. However, there was no clear relationship between pH values and bacterial counts in rumen or rectum, although the SER group had the lowest aerobic plate counts in both rumen and rectum. A lower aerobic plate count when goats fed sericea hay than those fed bermudagrass hay has already been reported [14]. A probable explanation could be that there is limited availability of nutrients than can be readily used by bacteria in the rumen and rectal bacteria. Fermentation of starch by bacteria in the cecum and colon produce volatile fatty acids that could reduce the pH of the colonic digesta and inhibit bacteria, including *E. coli* [42]. However, *E. coli* counts were not influenced by a diet high in CT, although a diet high in phlorotannins by inclusion of a brown seaweed extract has been reported to reduce *E. coli* counts in the gastrointestinal tracts of goat [43].

Acetic acid concentration was lower and isobutyrate and valerate concentrations were higher in TS goats compared to NTS goats; however, diet did not influence rumen VFAs in the current experiment. In contract, an earlier study reported that sericea hay-fed Spanish × Kiko goats had higher rumen pH due to lower production of total volatile fatty acids, a condition that favored *E. coli* and total coliform growth in the rumen, although such an effect was absent in the rectum [14]. The effects of high-CT diet on rumen VFA concentrations do not appear to be consistent across studies, probably because of the differences in quality and CT content of sericea varieties used in different studies.

### 4.5. Meat Quality Characteristics

*Longissimus dorsi* muscle initial and final pH values were not affected by diet or stress, which is in agreement with previous studies in goats [1,44]. The goats used in this study were 8 months of age. Age of goats may have a prominent effect on antemortem stress-induced glycogenolysis as muscle glycogen levels have been reported to be lower in younger goats compared to older goats [1]. These authors suggested that there may be preslaughter situations that may not affect muscle pH, despite inducing a significant muscle glycogen breakdown prior to slaughter. Muscle glycogen content was not analyzed in the present study. The a* values (redness) were lower in SER group compared with BG or BG-DW groups; however, the a* values in all three dietary groups in this study were lower than values observed in previous studies for goat loin chops [1,44]. The ultimate pH values in this study were greater than 6.0 in all animals, although it is not clear if lower a* values were related to muscle pH values as the differences in color were more prominent in the non-transported animals. Cooking loss and WBSF were also not influenced by ST or Diet. This is in agreement with previous reports that dietary changes do not have a significant effect on physicochemical properties of meat in goats [45].

## 5. Conclusions

Sericea diet may enhance anti-inflammatory properties because of higher levels of polyphenols present in the plant. Sericea-fed goats may have better capacity to combat stress because of either the beneficial effects of high CT content, low gastrointestinal parasite load, or both. Stress levels in goats increase during transportation and there is also evidence that glucocorticoid response is lower when goats have lower worm load as indicated by lower cortisol concentrations in the BG-DW group compared to BG or SER diet groups. Plasma metanephrine and normetanephrine may be better measures to study the effects of stress than epinephrine and norepinephrine in goats due to their very short half-life. Feeding sericea hay to goats can decrease microbial counts in the gastrointestinal tract, particularly in the rectum. Transportation stress can affect rumen VFA concentrations, although its implications for production traits are unclear from this study. Overall, feeding sericea lespedeza hay to goats may have beneficial effects such as enhanced antioxidant and anti-inflammatory properties and reduced gut microbial counts, in addition to the well-documented antiparasitic effect. However, ‘Serala’ sericea lespedeza diet did not improve meat quality characteristics in this study, and its effect on meat color needs further investigation.

## Figures and Tables

**Figure 1 animals-11-02857-f001:**
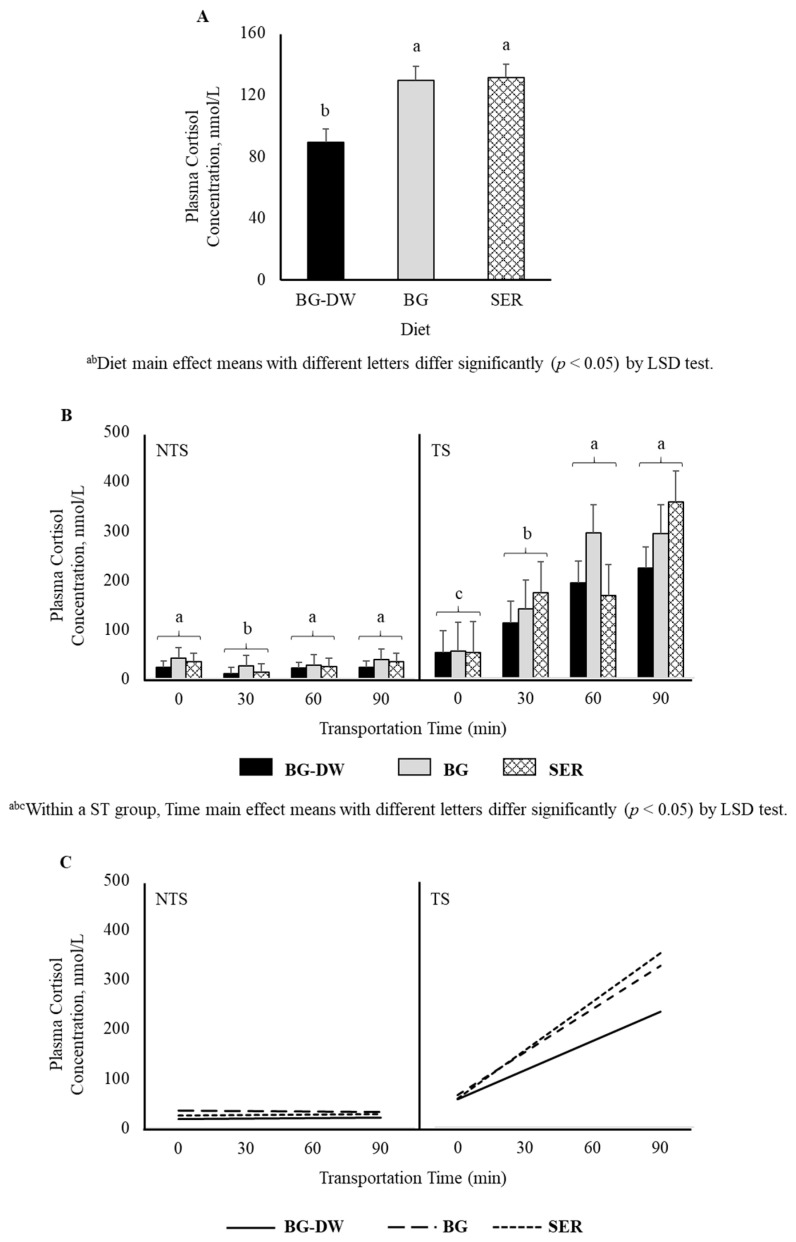
(**A**) Effects of Diet (BG-DW: bermudagrass hay—dewormed; BG: bermudagrass hay; SER: sericea hay; *p* < 0.05) and (**B**) Stress Treatment (ST; NTS: non-transported; TS: transported; *p* < 0.01) and Time (*p* < 0.01; ST × Time, *p* < 0.01) on plasma cortisol concentrations in goats. Bars represent means with SEM. (**C**) Polynomial contrasts showing a significant linear trend (*p* < 0.01) over Time.

**Figure 2 animals-11-02857-f002:**
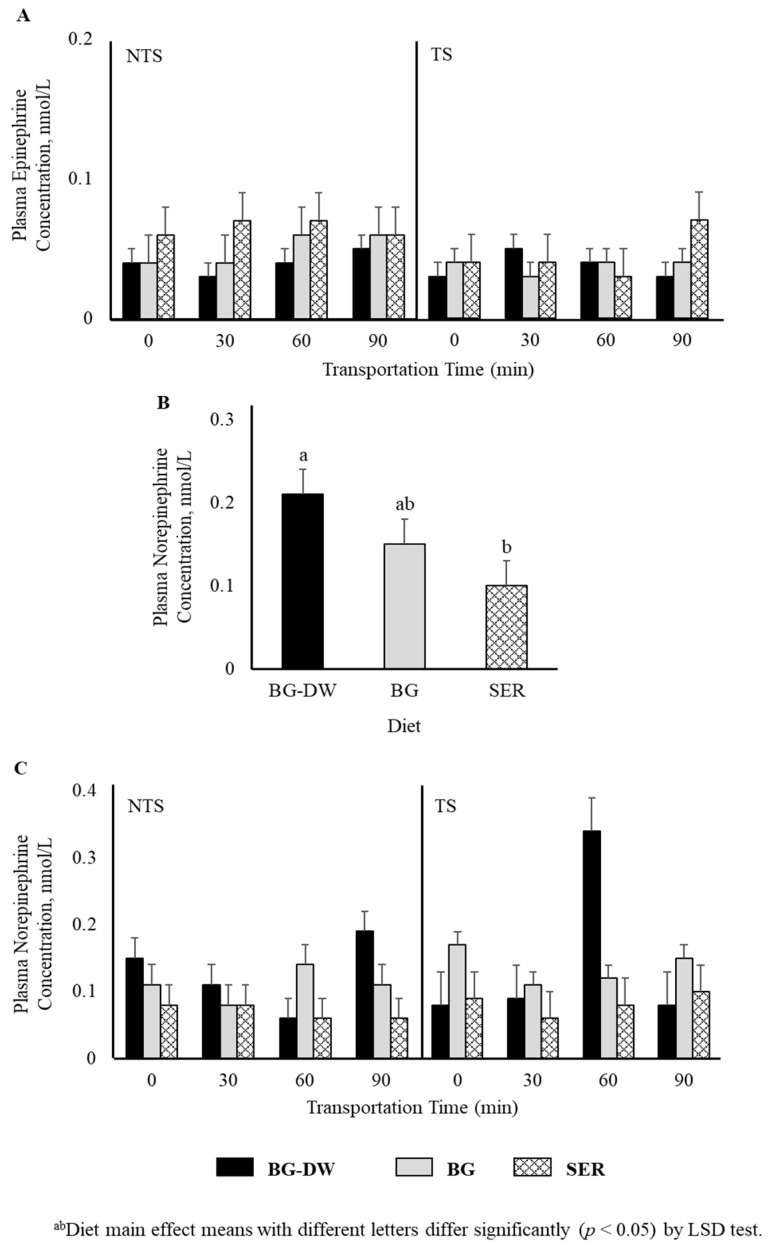
Effects of Diet (BG-DW: bermudagrass hay—dewormed; BG: bermudagrass hay; SER: sericea hay), Stress Treatment (ST; NTS: non-transported; TS: transported), and Time on plasma (**A**) epinephrine and (**B**,**C**) norepinephrine (Diet, *p* < 0.05; Diet × ST × Time, *p* < 0.01) concentrations in goats. Bars represent means with SEM.

**Figure 3 animals-11-02857-f003:**
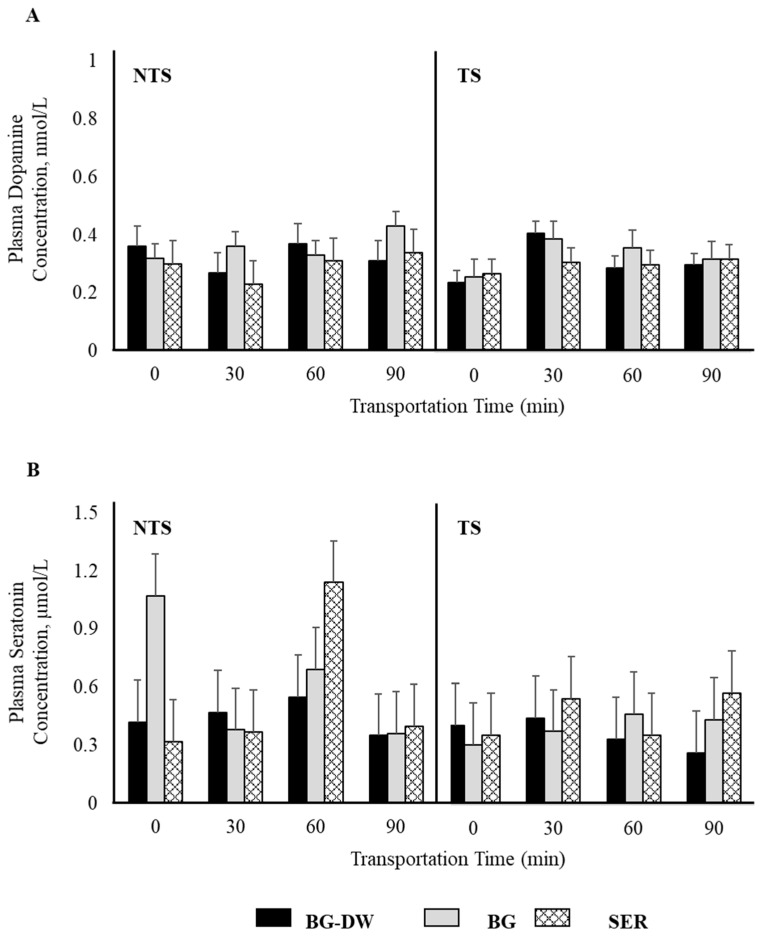
Effects of Diet (BG-DW: bermudagrass hay—dewormed; BG: bermudagrass hay; SER: sericea hay), Stress Treatment (ST; NTS: non-transported; TS: transported), and Time on plasma (**A**) dopamine (ST × Time, *p* = 0.07) and (**B**) serotonin (ST × Time, *p* < 0.05) concentrations in goats. Bars represent means with SEM.

**Figure 4 animals-11-02857-f004:**
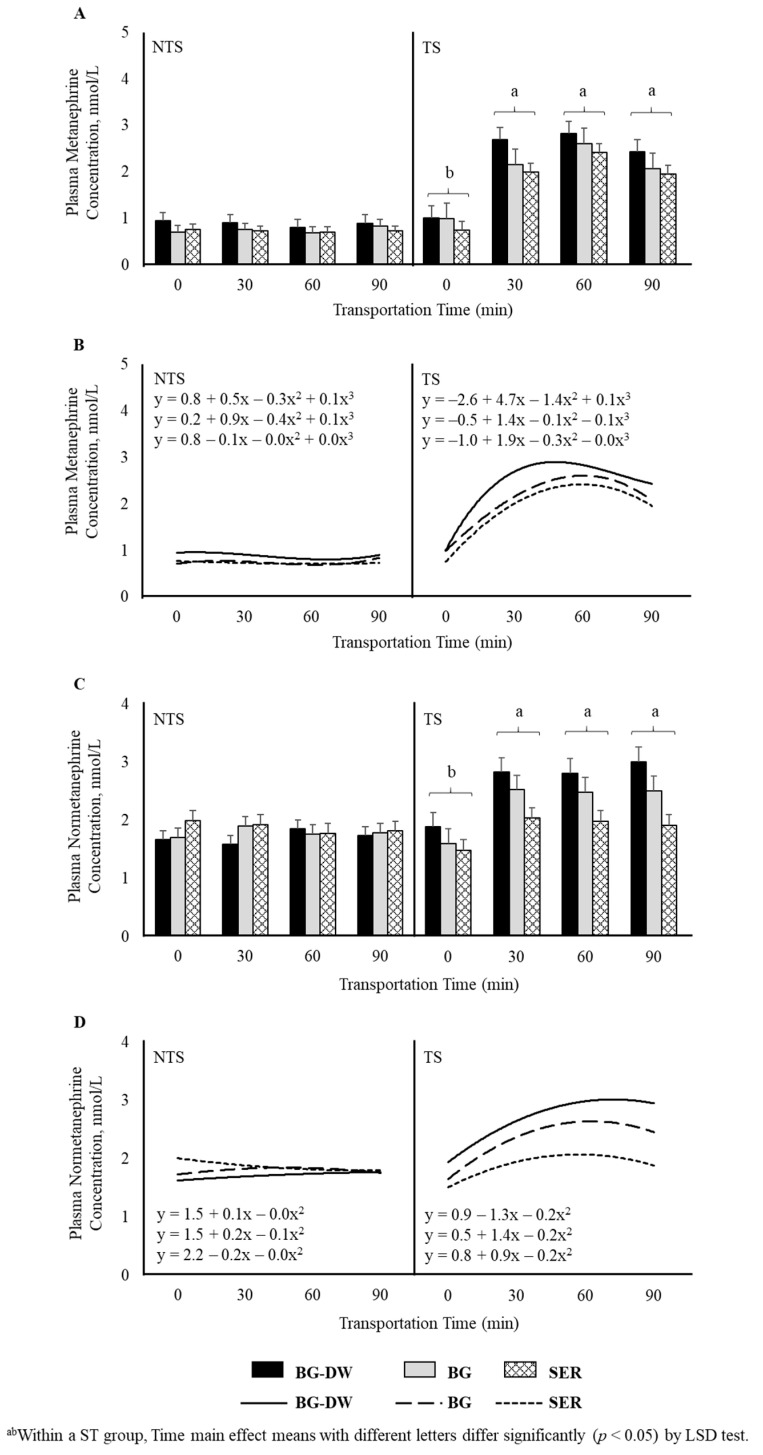
Effects of Diet (BG-DW: bermudagrass hay—dewormed; BG: bermudagrass hay; SER: sericea hay), Stress Treatment (ST; NTS: non-transported; TS: transported), and Time on plasma (**A**) metanephrine (ST, *p* < 0.01; Time, *p* < 0.01; ST × Time, *p* < 0.01) concentrations in goats. (**B**) Polynomial contrasts showing a significant cubic trend (*p* < 0.01) over Time. (**C**) Normetanephrine (ST, *p* < 0.01; Time, *p* < 0.01; ST × Time, *p* < 0.01; Diet × ST, *p* < 0.01) concentrations in goats. (**D**) Polynomial contrasts showing a significant quadratic trend (*p* < 0.01) over Time. Bars represent means with SEM.

**Figure 5 animals-11-02857-f005:**
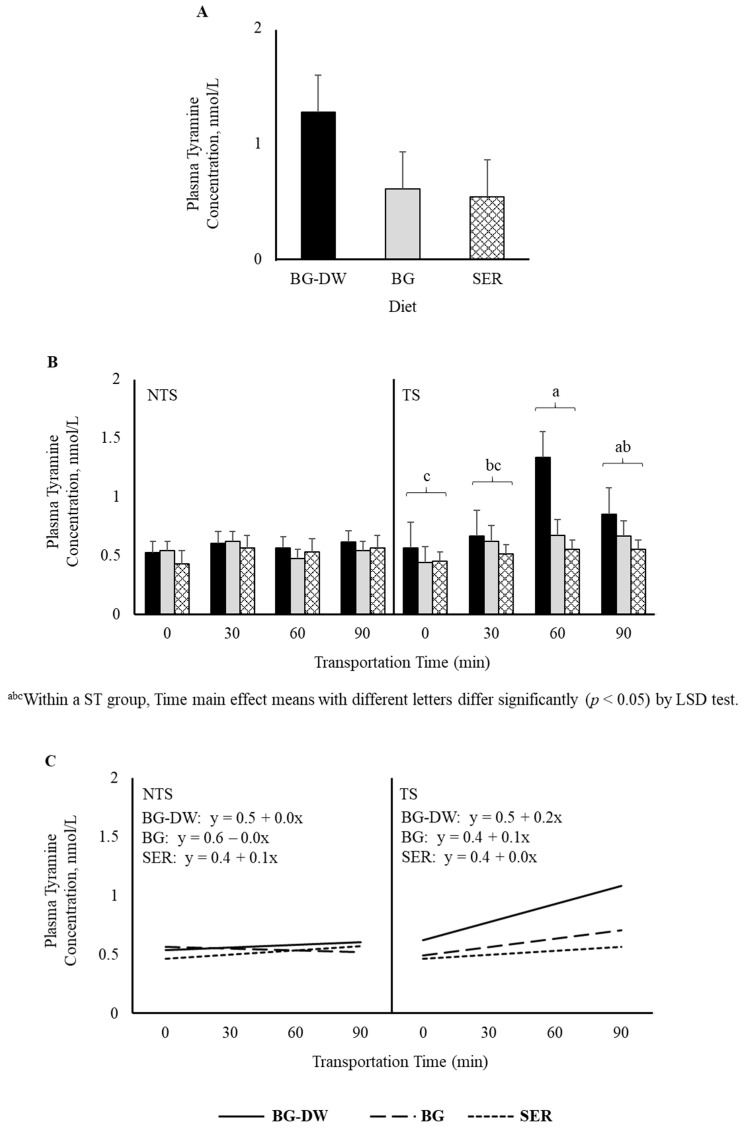
(**A**) Effects of Diet (BG-DW: bermudagrass hay—dewormed; BG: bermudagrass hay; SER: sericea hay; *p* = 0.07) and (**B**) Stress Treatment (ST; NTS: non-transported; TS: transported), and Time (Time, *p* < 0.05; ST × Time, *p* < 0.05) on plasma tyramine concentrations in goats. (**C**) Polynomial contrasts showing a significant linear trend (*p* < 0.01) over Time.

**Figure 6 animals-11-02857-f006:**
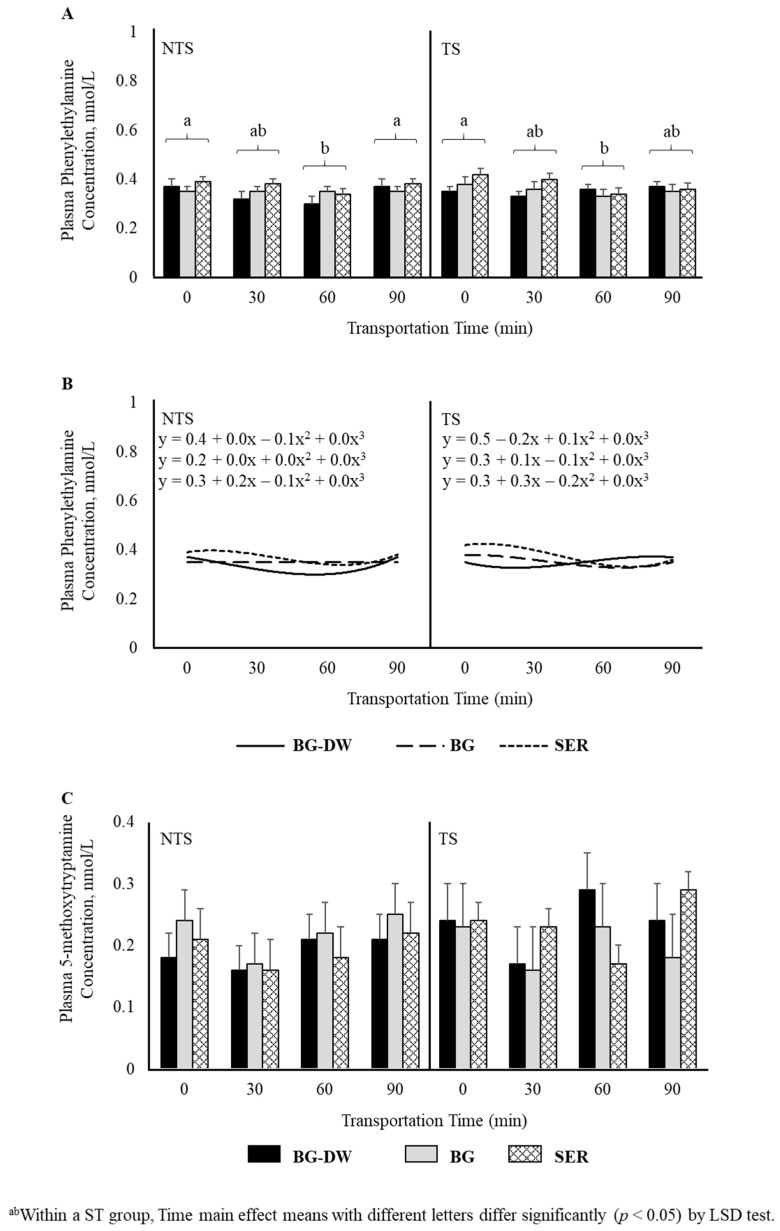
(**A**) Effects of Diet (BG-DW: bermudagrass hay—dewormed; BG: bermudagrass hay; SER: sericea hay), Stress Treatment (ST; NTS: non-transported; TS: transported), and Time (Time, *p* < 0.01) on plasma phenylethylamine concentrations in goats. (**B**) Polynomial contrasts showing a significant cubic trend (*p* < 0.05) over Time. (**C**) Plasma 5-methoxytryptamine concentrations in goats. Bars represent means with SEM.

**Figure 7 animals-11-02857-f007:**
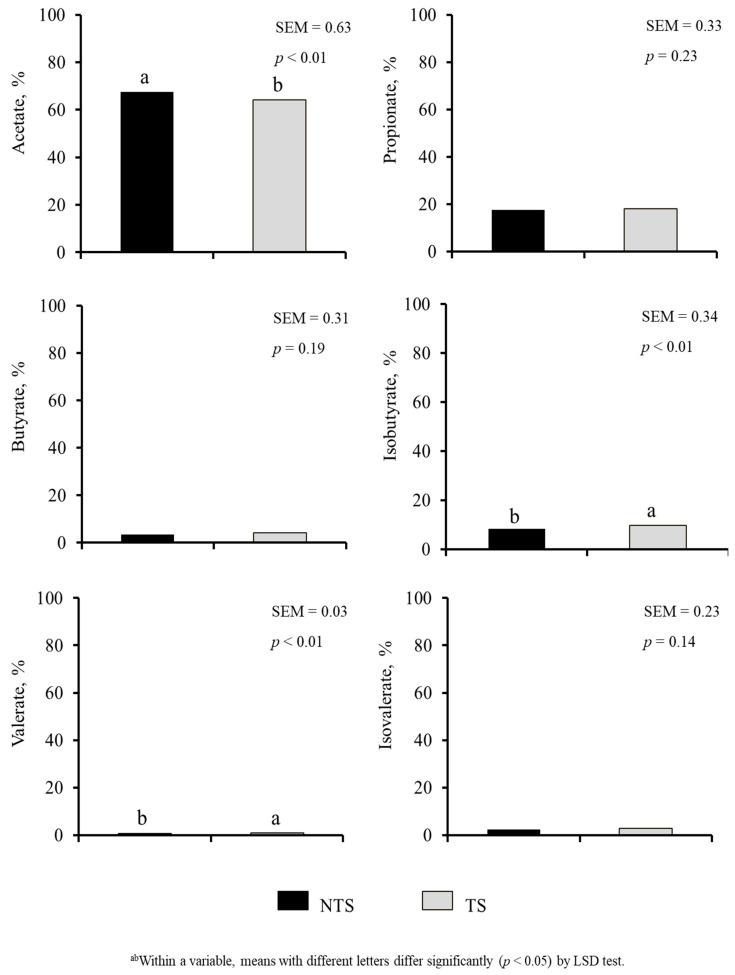
Effects of Stress Treatment (ST; NTS: non-transported; TS: transported) on rumen volatile fatty acid composition. Acetate, isobutyrate and valerate concentrations were significantly influenced by ST (*p* < 0.01).

**Table 1 animals-11-02857-t001:** Ingredients of the diets (hay + concentrate).

Ingredients, %.	Bermudagrass Diet (BG)	‘Serala’ Sericea Lespedeza Diet (SER)
Bermudagrass hay	75.0	0
‘Serala’ sericea lespedeza hay	0	75.0
Ground corn	6.0	6.0
Soybean meal	12.0	2.7
Poultry fat ^1^	1.6	1.6
Molasses (dry)	3.0	3.0
TM salt (red salt) ^2^	0.7	0.5
Vitamin Premix ^3^	0.7	0.5
Calcium carbonate	1.0	0
Biofos (mono calcium phosphate)	0	0.7

^1^ Mixed in the concentrate portion of the diet. ^2^ Contained >12% Zn, 10% Mn, 5% K, 2.5% Mg, 1.5% Cu, 0.3% I, 0.1% Co, and 0.02% Se. ^3^ Contained 2,000,000 IU of vitamin A, 400,000 IU of vitamin D3 and 230 IU of vitamin E/kg.

**Table 2 animals-11-02857-t002:** Pearson correlation coefficients between plasma cortisol and selected biogenic amine concentrations in each diet group (BG-DW: bermudagrass hay—dewormed; BG: bermudagrass hay; SER: sericea hay).

Item	Diet	Tyramine nmol/L	Dopamine nmol/L	Epinephrine nmol/L	Nor-Epinephrine nmol/L	Metanephrine nmol/L	Nor-Metanephrine nmol/L
Cortisol nmol/L	BG-DW	0.2261	0.0719	−0.1514	0.1777	0.6697	0.6273
		*p* = 0.1222	*p* = 0.6272	*p* = 0.3042	*p* = 0.2270	*p* < 0.0001	*p* < 0.0001
	BG	0.2988	−0.0012	−0.2754	0.0704	0.8505	0.5625
		*p* = 0.0391	*p* = 0.9936	*p* = 0.0581	*p* = 0.6342	*p* < 0.0001	*p* < 0.0001
	SER	0.1974	0.1478	−0.0938	−0.0339	0.7411	0.2027
		*p* = 0.1786	*p* = 0.3160	*p* = 0.5259	*p* = 0.8193	*p* < 0.0001	*p* = 0.1671

**Table 3 animals-11-02857-t003:** Effects of diet (BG-DW: bermudagrass hay—dewormed; BG: bermudagrass hay; SER: sericea hay) and stress treatment (ST: Non-Transported, NTS; Transported, TS) on differential leukocyte counts.

Leukocyte, %	Diet	ST	Time				n	*p*-Value by ANOVA				
			0 min		90 min			Diet	ST	Time	Diet × ST	ST × Time
			Mean	SEM	Mean	SEM						
Neutrophil	BG-DW	NTS	41.1	+2.44/−2.30	43.7 ^y^	+2.60/−2.45	6	0.609	0.001	0.001	0.048	0.001
		TS	36.5 ^b^	+2.17/−2.05	52.3 ^a,x^	+3.10/−2.93	6					
	BG	NTS	37.6	+2.23/−2.11	40.8 ^y^	+2.42/−2.29	6					
		TS	38.4 ^b^	+2.28/−2.15	59.5 ^a,x^	+3.54/−3.34	6					
	SER	NTS	39.2	+2.33/−2.20	40.9 ^y^	+2.43/−2.30	6					
		TS	39.7 ^b^	+2.36/−2.22	62.6 ^a,x^	+3.72/−3.51	6					
Lymphocyte	BG-DW	NTS	54.0	+3.19/−3.01	52.1 ^x^	+3.08/−2.90	6	0.176	0.001	0.001	0.011	0.001
		TS	58.9 ^a^	+3.48/−3.29	43.2 ^b,y^	+2.55/−2.41	6					
	BG	NTS	57.2	+3.38/−3.19	54.5 ^x^	+3.22/−3.04	6					
		TS	55.3 ^a^	+3.27/−3.08	33.9 ^b,y^	+2.00/−1.89	6					
	SER	NTS	56.2	+3.32/−3.13	54.3 ^x^	+3.21/−3.03	6					
		TS	55.1 ^a^	+3.25/−3.07	31.6 ^b,y^	+1.87/−1.76	6					
Monocyte	BG-DW	NTS	1.5	+0.28/−0.24	1.1	+0.21/−0.17	6	0.053	0.756	0.483	0.318	0.090
		TS	1.1	+0.21/−0.17	1.3	+0.23/−0.20	6					
	BG	NTS	1.7	+0.31/−0.26	1.3	+0.23/−0.20	6					
		TS	1.7	+0.31/−0.26	2.0	+0.38/−0.32	6					
	SER	NTS	1.7	+0.31/−0.26	1.5	+0.28/−0.24	6					
		TS	1.5	+0.28/−0.24	1.5	+0.28/−0.24	6					
Basophil	BG-DW	NTS	1.1	+0.13/−0.12	1.3	+0.15/−0.13	6	0.005	0.764	0.764	0.188	0.370
		TS	1.4	+0.17/−0.15	1.1	+0.17/−0.15	6					
	BG	NTS	1.3	+0.15/−0.13	1.3	+0.15/−0.13	6					
		TS	1.3	+0.15/−0.13	1.0	+0.12/−0.10	6					
	SER	NTS	1.0	+0.12/−0.10	1.0	+0.12/−0.10	6					
		TS	1.0	+0.12/−0.10	1.0	+0.12/−0.10	6					
Eosinophil	BG-DW	NTS	1.6	+0.30/−0.25	1.4	+0.26/−0.22	6	0.944	0.561	0.921	0.384	0.561
		TS	1.3	+0.24/−0.20	1.4	+0.26/−0.22	6					
	BG	NTS	1.5	+0.28/−0.24	1.3	+0.24/−0.20	6					
		TS	1.6	+0.30/−0.25	1.4	+0.26/−0.22	6					
	SER	NTS	1.3	+0.24/−0.20	1.4	+0.26/−0.22	6					
		TS	1.5	+0.28/−0.24	1.8	+0.33/−0.28	6					

^a,b^ Means within a row with different superscripts differ significantly (*p* < 0.05) by LSD test. ^x,y^ Means for a variable within a column with different superscripts differ significantly (*p* < 0.05) by LSD test.

**Table 4 animals-11-02857-t004:** Effects of diet (BG-DW: bermudagrass hay—dewormed; BG: bermudagrass hay; SER: sericea hay) and stress treatment (ST: Non-Transported, NTS; Transported, TS) on pH and microbial counts in the rumen and rectum of goats.

Item	ST	Diet			n	SEM	*p*-Value by ANOVA		
		BG-DW	BG	SER			Diet	ST	Diet × ST
*Rumen*									
pH	NTS	6.27	6.02	6.26	6	0.108	0.239	0.802	0.339
	TS	6.11	6.18	6.32	6				
*E. coli* log_10_ CFU/g	NTS	0.19	0.42	0.37	6	0.116	0.920	0.967	0.153
	TS	0.44	0.29	0.27	6				
Coliform log_10_ CFU/g	NTS	0.24	0.49	0.34	6	0.127	0.779	0.109	0.113
	TS	0.72	0.42	0.44	6				
Aerobic Plate log_10_ CFU/g	NTS	7.21	6.65	5.44	6	0.562	0.001	0.192	0.335
	TS	6.69 ^a^	6.84 ^a^	4.04 ^b^	6				
*Rectum*									
pH	NTS	6.69	6.79 ^x^	6.57	6	0.213	0.720	0.059	0.023
	TS	6.63	6.41 ^y^	6.63	6				
*E. coli* log_10_ CFU/g	NTS	4.58	4.77	4.60	6	0.405	0.771	0.227	0.671
	TS	5.39	4.95	4.80	6				
Coliform log_10_ CFU/g	NTS	3.43	4.19	4.35	6	0.351	0.056	0.201	0.398
	TS	4.13	4.03	4.88	6				
Aerobic Plate log_10_ CFU/g	NTS	6.76 ^ab^	7.20 ^a^	6.16 ^b^	6	0.195	0.001	0.087	0.133
	TS	6.74	7.33	6.90	6				

^a,b^ Means within a row with different superscripts differ significantly (*p* < 0.05) by LSD test. ^x,y^ Means for a variable within a column with different superscripts differ significantly (*p* < 0.05) by LSD test.

**Table 5 animals-11-02857-t005:** Effects of diet (BG-DW: bermudagrass hay—dewormed; BG: bermudagrass hay; SER: sericea hay) and stress treatment (ST: Non-Transported, NTS; Transported, TS) on *Longissimus* muscle pH, color, cooking loss, and Warner-Bratzler Shear Force (WBSF) value in goats.

Variable		ST	Diet			n	SEM	*p*-Value by ANOVA		
			BG-DW	BG	SER			Diet	ST	Diet × ST
Initial pH (15 min)		NTS	6.81	6.93	6.76	6	0.068	0.159	0.780	0.551
		TS	6.77	6.90	6.86	6				
Final pH (24 h)		NTS	6.27	6.02	6.26	6	0.108	0.239	0.802	0.339
		TS	6.11	6.18	6.32	6				
Color	L* value	NTS	46.45	44.63	45.77	6	2.043	0.185	0.279	0.174
		TS	50.02	49.24	43.10	6				
	a* value	NTS	7.52 ^ab^	7.63 ^a^	7.18 ^b^	6	0.102	0.001	0.130	0.832
		TS	7.41	7.43	7.09	6				
	b* value	NTS	7.23	8.07	7.94	6	0.966	0.585	0.945	0.837
		TS	7.22	7.45	8.42	6				
Cooking Loss, %		NTS	19.27	15.80	16.98	6	1.530	0.965	0.260	0.134
		TS	14.46	17.28	16.00	6				
WBSF, kg		NTS	3.75	3.03	3.09	6	0.403	0.510	0.215	0.671
		TS	3.79	3.78	3.56	6				

^a,b^ Means within a row with different superscripts differ significantly (*p* < 0.05) by LSD test.

## Data Availability

The data presented in this study are available on request from the corresponding author.

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
