# Peer review of "High-Condensed Tannin Diet and Transportation Stress in Goats: Effects on Physiological Responses, Gut Microbial Counts and Meat Quality"

_animals, 2021, doi:10.3390/ani11102857_

Round 1
Reviewer 1 Report
The research paper title “High-Condensed Tannin Diet and Preslaughter Stress in Goats: Effects on Physiological Responses, Gut Microbial Counts and Meat Quality” has been reviewed. This study explored the effects of dietary sericea lespedeza supplementation and transportation stress on physiological responses, gut microbial counts and meat quality of goats. Despite much interest for readers, there were many issues that are needed for addressing, and below are my concerns:
L2: Actually, it was transportation stress. The current title expressed intricately.
L14-16: Although transportation influenced physiological responses, stress or diet did not influence muscle pH, which is an important variable that affects the eating quality of meat. Did you mean transportation stress? This sentence makes me puzzled.
L21: Completely Randomized Design, why here should capitalize the first letter?
L41-44: Evidence from previous studies suggests that transportation stress elicits metabolic and physiological changes [3] that impact adrenal cortical activity, immune response, meat quality, and body weight in livestock [4]. How to understand such a long sentence with two “that”?
L45: Previous study showed 2h transportation to induce stress, how could you confirm the current 90 min is enough to induce stress?
L70-73: Put in a new paragraph.
L81-82: The abbreviation of each treatment group is complicated, for example, SL, BG, CON.
L104: Table 1: add notes for Vitamin premix, Poultry fat. Besides, there is no need to list Concentrate alone, because you have indicated the H:C ratio of 75:25 in L82.
L111: to minimize the effect of the order of blood sampling. What is the meaning of order here?
L113-114: Each animal was sampled four times (two from each side) during the 90-min treatment period. Why should here be samples four times and what is the exact time point during the 90 min?
L156: rumen liquor and rectal pH were determined using a pH meter. The pH of rumen content or just rumen fluid?
L176-177: Samples were frozen at 80 °C for ammonia nitrogen and volatile fatty acid analysis. Confirm if it was -80 °C.
L177-178: Rumen fluid samples were thawed overnight at 4 °C. How long was overnight?
L179-180: A subsample of the supernatant was removed for rumen ammonia assay. Please detail the method used for ammonia determination.
L188-189: The probe was inserted directly into the Longissimus dorsi muscle to determine pH. Here, Longissimus dorsi should be italic as Longissimus dorsi. Besides, have you determined the pH values 12/24/48 after skinning?
L199: with the samples being sheared at right angles to the orientation of muscle fibers. Please indicate the parameters for the determination of WBSF, i.e. speed.
L222: Figure 1B, the average with SEM seems too large and it is hard to introduce significant differences. I am wondering if you could provide me with the raw data for further assessment. The similar cases for Figure 2-6.
L325: The error bars should be added and value of vertical coordinates should be according to its exact values, not in the same.
L342-343: Diets high in tannins have shown to ameliorate the negative effects of stress in ruminant animals, in addition to reducing E. coli counts in their gastrointestinal tracts. I am not sure if this conclusion was from other literature or the current study.
L506: The conclusions seem too long and had not well- summarized the present work.
L539: For section of References, current form is inconsistent, please pay more attention to its style, journal name, page, doi, etc.
Reviewer 2 Report
This article investigated the effects of high-condensed tannin diet and transportation stress before stress on the on physiological responses, gut microbial counts and meat quality in goats. This research is interesting and helpful in antioxidant enhancement and anti-inflammatory properties during stress in meat goat production. The experiment was well designed and the data was collected correctly. But there were still some major problems in this manuscript.
Material and Methods:
Line 79-80, What was the month age of goats used in this research?
Line 81-87, the treatment was not clearly declared. What was the purpose to set the bermudagrass hay (BG) treatment?
Line 186: why did you test the initial pH instead of ultimate pH? Without the ultimate pH, we can not judge whether the lamb meat was the normal meat or DFD meat. You also found that the a* values were much lower, but do not know the reason. The ultimate pH would be useful to help explain this problem.
Line 190-192: how many chops were collected from each carcass? And why did you freeze the chop? Freezing and thawing will affect the cooking loss. How was the meat color tested? Was there a blooming before test? On frozen meat or meat after thawing?
Table 1: Why there was a difference in soybean meal and Calcium carbonate in the diet formulation between BG group and SER group? Would they also cause some different effects in the final results? Such as final meat quality and gut microbial diversity.
Line 204-211, for the blood data, I think they also can be analyzed as a split-unit design using GLM procedures.
Results:
Line 223-224: Why the BG and SER increased the plasma cortisol level?
Fig.1- Fig.5: When did the blood was collected for these data in Fig.1A, Fig.2B and Fig.5A? before transportation? or before slaughter? If the blood was collected just before the transportation, without the transportation stress, the data can be analyzed like this. But, if the blood was collected after the transportation, there were transportation stress effect. It would be incorrect to analyzed the data in this way. Please have a check.
Fig.1B, Fig.2A&C, Fig.3 A&B, Fig.4 A&C and Fig.5B: I think it is not a good way to analyzed the data and show the results like this. There was diet treatment, transportation treatment, transportation time and their interaction effects on these items. You need to do this analysis using split-unit design. So, please have a check and there probably would be different results.
Line 314-317: Was there no effect of diet on VFA?
Line 328-330: why there was a decrease in a* values in SER group?
Line 493: Longissimus dorsi should be in Italics.
Line 500-503, as a red meat, the a* values were surely lower the normal values. I can not find how did you test the meat color. How about the treatment of samples? Under VP package? Or after blooming? In frozen or after thawing?
As expected, the SER diet would improve the a* values because of its antioxidant ability, but the results were different, why?
Reviewer 3 Report
Review of the manuscript (ID: animals-1348243) entitled ‘High-Condensed Tannin Diet and Preslaughter Stress in Goats: Effects on Physiological Responses, Gut Microbial Counts and Meat Quality’ for Animals
Manuscript reports new insight on the improving goat’s welfare through preslaughter stress reduction using high condensed tannin diet. The authors conducted valuable investigations and made a very thorough statistical analysis of the collected research results.
The aims are clear, the observations based upon reliable methods, and the discussion is sounds. The manuscript will be acceptable for publication once the criticisms below have been addressed.
Conclusions should be more specific. Please, precise the implications.
My minor critics is also dealing with the non-compliance with the guideline for authors.
According to Animals Instruction for authors: ‘SI Units (International System of Units) should be used. Imperial, US customary and other units should be converted to SI units whenever possible.’ – please, convert conventional units of cortisol and catecholamines to SI units.
Latin (systematic names) like Sericea lespedeza, Longissimus dorsi, etc. should be written in Italics.
I suggest adding the units in headlines of tables – I know that they are placed in tables description, but the tables should speak for itself, with clear headlines.
There are numerous editorial errors in this manuscript. For example:
- Some words are highlighted (lines 5, 33).
- ‘and’ between Lee Thomas should be deleted (line 5)
- ‘Summary’ should be written in bold (line 10)
- The points in subtitles, e. g. ‘Materials and Methods’, ‘Animals and Diets’ are wrong placed
- References are not prepared according to the guideline.
Please, check the whole paper, these are only the examples.
Round 2
Reviewer 1 Report
The revised manuscript "High-Condensed Tannin Diet and Transportation Stress in Goats: Effects on Physiological Responses, Gut Microbial Counts and Meat Quality” has been reviewed. One major concern is the novelty and integrity of the current work. As the title indicated, the content of Tannin should be provided for each hay and diet. Besides, did it feasible in pratice by using these additives? Below are my point by point response to your reply.
L14-16: Although transportation influenced physiological responses, stress or diet did not influence muscle pH, which is an important variable that affects the eating quality of meat. Did you mean transportation stress? This sentence makes me puzzled.
Revised to improve clarity. Please see lines 14-16.
Comment: How could you say that transportation stress did not influence muscle pH? Are you sure?
L45: Previous study showed 2h transportation to induce stress, how could you confirm the current 90 min is enough to induce stress?
Previous reports have also indicated that stress in goats increases immediately after loading on to transport trailers and starts increasing thereafter. For example, one study showed that plasma cortisol concentrations increase within 30 min after beginning of transportation. This study is cited here in the revised version. Please see lines 43-45.
Comment: While, I am wondering which parameter(s) in your study proved the existed stress?
L70-73: Put in a new paragraph.
We feel a single sentence cannot be in a paragraph by itself. We prefer to leave it the way it is.
Comment: At least a conjunction should be provided before the objective!
L156: rumen liquor and rectal pH were determined using a pH meter. The pH of rumen content or just rumen fluid?
Revised. Rumen content. Please see line 162.
Comment: How you obtained the rumen fluids used for VFA analysis.
L177-178: Rumen fluid samples were thawed overnight at 4 °C. How long was overnight?
Approximately for 12 hours. Revised. Please see line 185.
Comment: This should not be Approximately in scientific experiment.
L539: For section of References, current form is inconsistent, please pay more attention to its style, journal name, page, doi, etc.
Revised according to the journal style.
Comment: Still not the way required.
